

# The influence of local oil exploration, regional wildfires, and long range transport on summer 2015 aerosol over the North Slope of Alaska

Jessie M. Creamean[1,2], Maximilian Maahn[1,2], Gijs de Boer[1,2], Allison McComiskey[3], Arthur J. Sedlacek[4], Yan Feng[5]

[1]Cooperative Institute for Research in Environmental Sciences, University of Colorado, Boulder, CO, USA
[2]Physical Sciences Division, National Oceanic and Atmospheric Administration, Boulder, CO, USA
[3]Global Monitoring Division, National Oceanic and Atmospheric Administration, Boulder, CO, USA
[4]Environmental and Climate Sciences, Brookhaven National Laboratory, Upton, NY, USA
[5]Environment Science Division, Argonne National Laboratory, Lemont, IL, USA

*Correspondence to*: Jessie M. Creamean (jessie.creamean@noaa.gov)

**Abstract.** The Arctic is warming at an alarming rate, yet the processes that contribute to enhanced warming are not well understood. Arctic aerosols have been targeted in studies for decades due to their consequential impacts on the energy budget directly and indirectly through their ability to modulate cloud microphysics. Even with the breadth of knowledge afforded from these previous studies, aerosols and their effects remain poorly quantified, especially in the rapidly-changing Arctic. Additionally, many previous studies involved use of ground-based measurements, and due to the frequent stratified nature of the Arctic atmosphere, brings into question the representativeness of these datasets aloft. Here, we report on airborne observations from the U.S. Department of Energy Atmospheric Radiation Measurement (ARM) program's Fifth Airborne Carbon Measurements (ACME-V) campaign along the North Slope of Alaska during the summer of 2015. Contrary to previous evidence that the Alaskan Arctic summertime air is relatively pristine, we show how local oil extraction activities, 2015's central Alaskan wildfires, and to a lesser extent, long-range transport introduce aerosols and trace gases higher in concentration than previously reported in Arctic haze measurements to the North Slope. Although these sources were either episodic or localized, they serve as abundant aerosol sources that have the potential to impact a larger spatial scale after emission.

## 1 Introduction

The Arctic climate is warming rapidly compared to other locations globally, which has implications for anomalous snow and ice melt (Jeffries et al., 2013). Replacement of highly reflective surfaces by darker, more absorbing surfaces (i.e., tundra and open ocean water) enhances Arctic warming, especially in the summer (Chapin et al., 2005). Such warming subsequently impacts the ecological cycle, socioeconomics, and midlatitude weather patterns (Screen and Simmonds, 2010; Serreze and Barry, 2011). This warming feedback is one in a complex system of interrelated processes resulting in "Arctic



amplification", the observed enhanced warming seen in the Arctic to date, and in part motivates the need to improve our understanding of atmospheric processes that modulate energy reaching the Arctic surface (Serreze and Francis, 2006).

In addition to climate feedback described above, the principal atmospheric constituents that perturb the surface energy budget, beyond greenhouse gases, are clouds and aerosols. Aerosols can directly scatter and absorb solar radiation or indirectly impact atmospheric radiation through their role in cloud lifecycle, by serving as cloud condensation nuclei (CCN) and/or ice nucleating particles (INPs). However, the ability of aerosols to serve as CCN or INPs hinges on their composition, size, and number concentration, each of which inherently depends on their source and atmospheric processing. Predominant sources of Arctic aerosols have been examined by a wide spectrum of previous aerosol characterization studies, including numerous ground-based and airborne research campaigns conducted in the Alaskan Arctic, extending back to the mid-20[th] century (e.g., Schnell and Raatz, 1984; Barrie, 1986; Delene and Ogren, 2002; Verlinde et al., 2007; Quinn et al., 2009; Brock et al., 2011; McFarquhar et al., 2011). To better understand aerosol properties in this environment, two atmospheric research facilities have been established on the North Slope of Alaska that encompass routine, aerosol measurements—including, but not limited to, aerosol optical, physical, and chemical properties, and CCN concentrations.

Utqiaġvik, Alaska (formally Barrow) features an observatory established by the National Oceanic and Atmospheric Administration (NOAA) Earth System Research Laboratory's (ESRL) Global Monitoring Division (GMD) in 1976. Several previous studies have focused on different combinations of the long-term, ground-based aerosol optical, physical, and/or chemical property measurements to evaluate the annual cycle of aerosol sources at Utqiaġvik (e.g., Polissar et al., 2001; Delene and Ogren, 2002; Quinn et al., 2002; Quinn et al., 2009). For example, Quinn and colleagues used aerosol number concentrations, optical properties, and chemistry measurements to conclude that the winter and spring are impacted by aerosol transported from mid-latitudes, while summer and fall aerosols contain contributions from local biological activity, sea salt, and/or residual (i.e., unanalysed) aerosol mass that may represent mineral or organic species (Quinn et al., 2002; Quinn et al., 2009). More recently, Kolesar and colleagues (2017) used a 6-year time series of particle size distributions to discover that particle growth events occurring at Utqiaġvik resulted from emissions originating from the oil and gas fields of the Prudhoe Bay area, approximately 200 miles east of Utqiaġvik. Gunsch and colleagues (2017) found submicron (i.e., < 1 μm in diameter) combustion-derived particles were transported from the Prudhoe Bay oil field to Utqiaġvik 10% of the time during their Aug – Sep 2015 study.

In addition to Utqiaġvik, another Northern Alaskan facility was recently established by the U.S. Department of Energy (DOE) Atmospheric Radiation Measurement (ARM; since 2013) program at Oliktok Point (https://dis.arm.gov/sites/amf/oli/). This site exists on the north-westerly edge of oil extraction activities in Prudhoe Bay. Aerosol optical, physical, and chemical property measurements were implemented during the summer of 2016. This site provides the ability to determine the potential impacts of oil extraction emissions on the relatively pristine Arctic atmosphere. Overall, the North Slope provides a unique opportunity to investigate aerosols and their impacts from the clean Arctic background, long-range transported aerosol from lower latitudes, and regional oil extraction activities.



While the studies discussed in the previous paragraphs have provided significant insight into aerosol properties in Northern Alaska, one crucial deficiency is they comprise *only* ground-based observations of aerosol. The Arctic atmosphere can be highly stratified, thus disconnecting aerosol measured at the surface from those aloft, including at the height at which cloud modulation by aerosols occurs. To bridge this gap, numerous airborne campaigns have focused on evaluating Alaskan

Arctic aerosol sources and aerosol-cloud interactions. For example, during the March 1983 NOAA Arctic Gas and Aerosol Sampling Program (AGASP) flights over Alaska, the highest aerosol concentrations were found from the mid- to upper-troposphere (Schnell and Raatz, 1984). Several airborne campaigns—including Aerosol, Radiation, and Cloud Processes affecting Arctic Climate (ARCPAC), Arctic Research of the Composition of the Troposphere from Aircraft and Satellites (ARCTAS-A), and Indirect and Semi-direct Aerosol Campaign (ISDAC)—took place in the region during April 2008 to

characterize tropospheric pollution and its sources during the Arctic haze season during the International Polar Year. These flights aimed to improve our knowledge on how changes in aerosol composition and concentration directly impact atmospheric radiation, while others extended beyond this to examine how aerosols influence cloud properties and their associated radiative forcing (Brock et al., 2011; McFarquhar et al., 2011; Bian et al., 2013). These studies presented valuable information on the vertical structure of Arctic aerosol, and the relative contributions from Arctic background, fossil fuels,

and biomass burning sources, but are limited to April or March only. Airborne measurements available from the Mixed-Phase Arctic Cloud Experiment (M-PACE), which took place from late September to late October 2008, are predominantly focused on clouds and, with respect to aerosols, only encompassed aerosol size distribution measurements and INP concentrations (Verlinde et al., 2007; Prenni et al., 2009; Jackson et al., 2012). To our knowledge, only one study reports airborne *in situ* aerosol measurements over the Alaskan Arctic during the summer: NASA's Arctic Boundary Layer

Experiment (ABLE 3A) during the summer of 1988 (Gregory et al., 1992). However, this study was limited to flights between Fairbanks and Utqiaġvik, and to aerosol size distributions from 0.12 to 8 µm in diameter—no other aerosol measurements were obtained.

With increasing prospects for (1) oil extraction, (2) added shipping routes due to a reduction in sea ice extent, and (3) warming temperatures inducing wildfires in sub-Arctic boreal regions, regional fossil fuel and biomass burning combustion

sources will continue to be of great importance to the aerosol population (Randerson et al., 2006; Gautier et al., 2009; Harsem et al., 2011; Peters et al., 2011; de Groot et al., 2013). However, local pollution and other high-latitude Eurasian resource extraction sources and their resulting impacts on clouds and radiation are poorly quantified (Arnold et al., 2016). Hobbs and Rango (1998) documented increased cloud droplet number concentrations in air masses originating around Prudhoe Bay through airborne measurements over the Beaufort Sea. In a companion paper by Maahn and colleagues (2017),

local emissions from Prudhoe Bay are shown to impact cloud drop size in comparison with more pristine clouds over Utqiaġvik. Such studies support the idea that emissions from oil extraction activities in this region have air quality and climatic implications and are important to assess. Additionally, Stohl and colleagues (2013) reported that gas flaring emissions are underestimated in the Arctic, further justifying the need to evaluate emissions from these sources.



In addition to industrial sources, it is recognized that Alaskan boreal fires periodically impact the aerosol population over the North Slope. Eck and colleagues (2009) reported high summer time (August) fire counts, impacting aerosol optical depths (AODs) over Utqiaġvik. Stohl and colleagues (2006) reported similar findings using ground-based absorption and CO (i.e., a tracer for biomass burning) measurements at Utqiaġvik. Both studies concluded that individual smoke transport

events resulted in short episodes of higher AOD and absorption values than typical springtime Arctic haze. Regardless of their episodic behaviour, summertime sub-Arctic boreal fires can produce substantial quantities of aerosol that can reside in the troposphere for 1 – 2 weeks (Stohl et al., 2013). The Arctic summertime atmosphere is predominantly decoupled from midlatitude sources and thus less polluted as compared to the rest of the year (Quinn et al., 2002; Leaitch et al., 2013; Heintzenberg et al., 2015), thus it is critical to assess the impacts of potentially important sources of aerosol on Arctic

radiation and cloud microphysical processes. Here, we present airborne aerosol and trace gas observations from the ARM ACME-V field campaign during the summer of 2015 to evaluate such sources in the Alaskan Arctic.

## 2 Methods

### 2.1 Study location and dates

ACME-V flights were conducted over the North Slope of Alaska between five waypoints, including Oliktok Point (70.51°N,

149.86°W), Utqiaġvik (71.29°N, 156.79°W), Atqasuk (70.48°N, 157.42°W), Ivotuk (68.49°N, 155,75°W), and Toolik Lake (68.63° N, 149.61° W) (Figure 1), all north of the Brooks Mountain Range. The campaign involved 38 research flights from 4 Jun to 9 Sep 2015, generally flying every 2 – 3 days (Table 1). The DOE ARM Gulfstream-1 (G-1; part of the ARM Aerial Facility) aircraft typically flew below 1000 m above ground level (m AGL) between the waypoints, while spiraling up to 8,000 m AGL above each waypoint. Flight tracks varied in the number and order of waypoints that were overflown.

**2.2 Aircraft aerosol and trace gas payload**

The G-1 was equipped with a suite of atmospheric state, cloud, aerosol, and trace gas instruments, (see https://www.arm.gov/research/campaigns/aaf2014armacmev for a complete list of instrumentation and available data) (Biraud et al., 2016), though in the current study we only focus on the aerosol and CO measurements. Total number concentrations (CN) of aerosol particles 3 nm – 3 µm and 10 nm – 3 µm in diameter were measured using two Condensation

Particle Counters (CPC, TSI, Inc. models 3025 and 3010, respectively). The CPC 3025 and 3010 have a 50% counting efficiency of 3-nm and 10-nm particles, respectively. Aerosol size distributions were measured using three different instruments, including an Ultra-High Sensitivity Aerosol Sizer (UHSAS, Droplet Measurement Technologies, Inc.), a Passive Cavity Aerosol Spectrometer (PCASP, Droplet Measurement Technologies, Inc. model SPP-200), and an Optical Particle Counter (OPC, Climet model C1-3100) in combination with a Multi Chanel Analyzer (Ortec model Easy-MCA-8k),

which measure particle optical diameters in the ranges of 0.06 – 1 µm, 0.1 – 3 µm, and 0.8 – 15 µm, respectively. The PCASP was operated with an anti-ice heater, thus the particles measured are predominantly dry (Kassianov et al., 2015). The



UHSAS experienced instrumental complications during most of the campaign, thus is not used for the current study to alleviate any limitations and skewness from operation dates. Total aerosol light scattering and absorption coefficients (Mm$^{-1}$) were measured using a 3-wavelength (450 nm, 550 nm, and 700 nm) nephelometer (TSI, Inc. model 3563) and 3-wavelenth (464 nm, 528 nm, and 648 nm) Particle Soot Absorption Photometer (PSAP, Radiance Research, Inc.), respectively.

Refractory black carbon (rBC) concentrations were measured with the Single Particle Soot Photometer (SP2, Droplet Measurement Technologies, Inc.), which measures individual rBC particles through laser-induced incandescence, making it selective for rBC (Sedlacek, 2016). Quality Assurance/Quality Control (QA/QC) checks of the SP2 data ensure that other potentially refractive particles such as mineral dust are not counted as rBC particles. Carbon monoxide (CO) concentrations were measured with a $CO/N_2O/H_2O$ instrument (Los Gatos Integrated Cavity Output Spectroscopy instrument model 907-

0015-0001) and is used as a tracer for combustion sources, including both biomass burning and fossil fuel (Andreae and Merlet, 2001; Brock et al., 2011; Liu et al., 2014). Carbon dioxide ($CO_2$) concentrations were measured by Cavity Ring Down Spectroscopy (Picarro model G2301) and together with the CO measurements were used to calculate Modified combustion efficiency (MCE) (Liu et al., 2014; Biraud and Reichl, 2016).

All data were collected at 1-second intervals and are publicly-available on the ARM data archive

(http://www.archive.arm.gov/armlogin/login.jsp). Data quality was verified through quality assurance and data quality checks by DOE ARM. CPC, PCASP, OPC, and rBC data flagged for being in-cloud were excluded from the current analysis, since the isokinetic inlet used on the G-1 during the study does not discern between interstitial aerosols and cloud particles, and cloud and aerosol size ranges can potentially overlap. Data periods impacted by liquid and ice clouds were defined as those having 1 x 10$^7$ m$^{-3}$ droplets and 100 m$^{-3}$ ice particles larger than 400 μm, respectively (Lance et al., 2011).

When a cloud was found (defined as at least 10 seconds of data where the cloud threshold is exceeded), aerosol observations 3 seconds before and 3 seconds after the cloud are discarded as well to avoid contamination of the aerosol probes with cloud particles (Maahn et al., 2017). CO data were used in- and out-of-cloud since there are not potential artefact issues. To minimize the influence of localized contamination from take-off and landing the Deadhorse airport (19.5 m AMSL), all data below 20 m AMSL and within 3 km of the airport were removed. All data are presented in Coordinated Universal Time

(UTC).

### 2.3 Aerosol dispersion modelling

Aerosol dispersion simulations were executed to demonstrate aerosol transport using version 4 of the Hybrid Single Particle Lagrangian Integrated Trajectory (HYSPLIT4) model (Draxler, 1999; Stein et al., 2015) and 1° data from the NOAA/NCEP Global Data Assimilation System (GDAS) (Kalnay et al., 1996). Simulation parameterization details are presented by Maahn

and colleagues (2017), but are reiterated briefly here. Aerosol mass concentrations were evaluated qualitatively from one central Prudhoe Bay location and from five locations within the active fire region in central Alaska at 100-m intervals from 0 to 5,000 m AGL for 72 hours, a 6-hour release of particles at a default emission rate of one arbitrary mass unit for the study time period (1 Jun – 30 Sep 2015). Other set parameters include particle density (6 g cm$^{-3}$), shape factor (1.0), particle



diameter (0.2 µm) (Eck et al., 1999; Rissler et al., 2006; Brock et al., 2011; Sakamoto et al., 2015), dry deposition velocity (1 x $10^{-4}$ m s$^{-1}$) (Warneck, 1999), in-cloud scavenging defined as a ratio of the pollutant in rain (g L$^{-1}$) measured at the ground to that in air (g L$^{-1}$ of air in the cloud layer) (4 x $10^4$), and below-cloud scavenging (5 x $10^{-6}$ s$^{-1}$). Radioactive decay and pollutant resuspension were set to the default values of zero days and 0 m$^{-1}$, respectively.

## 2.4 Supporting satellite data

The source of aerosols from the central Alaskan fires was determined using imagery from the Moderate Resolution Imaging Spectroradiometer (MODIS) on board the Terra satellite. MODIS Aqua looked similar, thus only Terra observations are discussed herein. Aerosol optical depth (AOD) data from MODIS were acquired from the Giovanni data server (http://giovanni.gsfc.nasa.gov/giovanni/) for daily dark target deep blue combined mean AOD at 550 nm and a 1° spatial resolution using a domain of 139°W to 169°W and 57°N to 72°N (MOD08_D3_6) (Ackerman et al., 1998). Fire and surface thermal anomaly data were also acquired from MODIS using brightness temperature measurements in the 4-µm and 11-µm channels (Giglio, 2010). The fire detection strategy is based on absolute detection of a fire (when the fire strength is sufficient to detect), and on detection relative to its background (to account for variability of the surface temperature and reflection by sunlight) (Giglio et al., 2003). The algorithms include masking of clouds, bright surfaces, glint, and other potential false alarms (Giglio et al., 2003). Swaths from overpasses over the state of Alaska were used to determine the daily locations of fires. The Alaskan fire season was relatively active (i.e., had the highest density of fires) from mid-Jun to mid-Jul 2015 (de Boer et al., 2017). Detected thermal anomalies or fires for the 4 Jun – 31 Aug period are used (data are not available from 1 – 9 Sep).

## 3 Results & discussion

### 3.1 Prudhoe Bay is a persistent local source of small particles in the boundary layer

Measurements from ACME-V below 500 m show a clear source of aerosol from Prudhoe Bay (Figure 2a, b), which was predominantly composed of particles with diameters ($D_p$) between 3 and 10 nm (CPC$_{diff}$; calculated from subtracting the CPC 3010 from CPC 3025 number concentrations). "Hot spots" of larger particles measured by the PCASP (0.1 – 3 µm) and OPC (0.8 – 15 µm) were not observed near Prudhoe Bay (not shown). The highest number concentrations of 3 – 10 nm particles (up to $10^4$ particles cm$^{-3}$) were observed within 50 km of Deadhorse (i.e., used here as a proxy for Prudhoe Bay; see Figure 1) in a layer from 100 to 300 m AMSL and during almost all flights (Figure 3). Particles within this size range are associated with nucleated aerosol (i.e., spontaneous *in situ* aerosol formation from precursor gases) (Colbeck and Lazaridid, 2014). These high number concentrations of small particles are likely formed from gas-to-particle partitioning of reactive gases from flaring and venting along the North Slope. Flaring and venting of gas, which is prominent near the surface in the Arctic near oil and gas facilities (Jaffe et al., 1995; Johnson and Coderre, 2011), could contribute the vapours—such as ozone, SO$_x$ and various aromatic hydrocarbons—that induce nucleation of new particles (Wilson and McMurry, 1981;





Parungo et al., 1992; Kulmala et al., 2004; Laaksonen et al., 2008; Ismail and Umukoro, 2012; Emam, 2015). The highest number concentrations of 3 – 10 nm particles were predominantly observed below 500 – 600 m, indicating: 1) removal of these particles via growth into the accumulation mode (0.1 – 2.5 µm) as the plume evolves and disperses vertically (Colbeck and Lazaridid, 2014) and/or 2) dynamical restriction of these particles in the atmospheric boundary layer. When examining the ratio of the $CPC_{diff}$ (i.e., nucleation mode) to PCASP number concentrations (i.e., accumulation mode) with altitude (Figure 4), the ratio is > 1 (i.e., nucleation mode particles were dominant) for 74% of the time and < 1 (i.e., accumulation mode particles were dominant) for 26% of the time below 500 m AMSL. Furthermore, this ratio decreased overall with altitude and with decreasing latitude (not shown), indicating the nucleation mode particles were formed at the lowest altitudes closest to their source, while accumulation mode particles originated from growth of the nucleation mode aerosol and/or a different source (see section 3.3).

Black carbon (as rBC) also had relatively high mass concentrations (up to 464 ng kg$^{-1}$) in the Prudhoe Bay area below 500 m AMSL (Figure 2c), which likely originates from local combustion sources such as on- and off-road vehicles, facility heating, and to some extent, flaring (Bond et al., 2013). Particles measured immediately near oil combustion sources—including rBC—normally have a size mode around 100 – 130 nm (Parungo et al., 1992; Chang et al., 2004). The smallest average mass equivalent modal size of the rBC from ACME-V were 115 nm and 110 nm for all data and those closest to Deadhorse (below 500 m AMSL nearest to and < 50 km from Deadhorse), respectively, indicating: 1) particles were "fresher" (i.e., less coated or "aged" from heterogeneous reactions) closest to the Prudhoe Bay source (Maahn et al., 2017) and 2) modal sizes are analogous to what might be expected from oil combustion sources after slight aging due to farther proximity from sources (i.e., not direct measurement from stacks). However, the larger rBC sizes further south of Deadhorse are likely influenced by biomass burning emissions as discussed in section 3.2.

Aerosols originating from the oil extraction activities were localized to the Prudhoe Bay area as suggested by the HYSPLIT dispersion model (Figure 2d). This reflects the source and physical removal processes occurring with the dominant-sized particles in this region. Both nucleation and Aitken mode (i.e., 10 – 100 nm) aerosols have lifetimes on the order of minutes to hours and thus have typical travel distances of 1 – 10s of km (Wilson and Suh, 1997). Such distances corroborate our finding that nucleation mode aerosol and rBC are restricted to < 50 km from Deadhorse below 500 m AMSL. Our results and those presented by Maahn and colleagues (2017) demonstrate an increase in aerosol size with distance from Prudhoe Bay, indicating loss to the accumulation mode.

Our results demonstrate that Prudhoe Bay is a strong and persistent source of nucleated aerosol and primary combustion aerosol, however, the high mass and number concentrations of these aerosols are restricted to the boundary layer and tend to remain localized to the Prudhoe Bay area. These aerosols may not have strong direct effects on the regional atmospheric radiation budget due to their inherently small size and low concentrations of larger accumulation mode particles (Friedlander, 2000). This is supported by the fact that no noticeable spatial patterns in absorption and scattering were observed as a function of distance from Deadhorse (not shown). However, as these particles age through atmospheric processing from co-emitted gases such as $SO_x/NO_x$ and grow larger into the accumulation mode, it is possible they could





have an impact after sufficient atmospheric residence time downwind. We do not have the compositional data necessary to determine the mixing state or extent of aging of these nucleation mode particles into the accumulation mode. Additionally, modeling studies have suggested that black carbon aerosols from Prudhoe Bay oil extraction have a positive net radiative forcing, particularly in the summer due to greater absorption of solar radiation (Ødemark et al., 2012). In terms of indirect

forcing, Maahn and colleagues (2017) demonstrated the importance of Prudhoe Bay industrial aerosol in modulation of cloud properties over the North Slope. Leaitch and colleagues (2016) and Burkhart and colleagues (2017) recently published observations of CCN down to 20 nm in the Canadian Arctic; contrary to the conventional wisdom that 100 nm is the threshold relevant for CCN. However, these Canadian Arctic aerosols were likely compositionally different due to their marine origin, and thus could vary in hygroscopicity as compared to oil extraction emissions. In general, our results show

that aerosols originating from Prudhoe Bay, although localized, could serve as a crucial source of aerosol that may impact radiation and clouds along the North Slope.

## 3.2 Regional fires impact air composition over much of Central and Northern Alaska

Another dominant aerosol source observed during the ACME-V field campaign was the central Alaskan wildfires. The summer 2015 season was particularly active, with fires detected via satellite during the entire study, and the highest density

of fires from mid-Jun to mid-Jul (Figure 5) (de Boer et al., 2017). These fires produced dense plumes of aerosol that propagated over much of the North Slope as evidenced by the elevated AOD originating from central Alaska. ACME-V flights were not significantly impacted by the high AOD regions until late-Jun and extended until the end of Jul. In particular, the G1 flew through the wildfire plumes 25 Jun – 1 Jul near Toolik Lake, 9 – 15 Jul over most of the flight track, and 16 – 22 Jul near Utqiaġvik and Oliktok Point.

*In situ* measurements from ACME-V show clear influence of Alaskan boreal fires (Figure 6). PCASP number, rBC mass, and CO were high in concentration, particularly at the southern portion of the flight track close to the Brooks Range. OPC concentrations, absorption coefficients, and scattering coefficients were also elevated in this region (not shown), but predominantly nearest to Toolik Lake as discussed in more detail below. Additionally, the HYSPLIT dispersion simulations from the five fire source points indicate larger particle mass concentrations spread over the same region as the high

concentrations of the *in situ* measurements (Figure 6d). Wildfires emit large quantities of primary organic aerosol (POA) and can generate secondary organic aerosol (SOA) that can be transported over long distances and topographical features (Andreae and Merlet, 2001; Collier et al., 2016; Creamean et al., 2016). Therefore, we would expect to observe an abundance of coarse and accumulation mode aerosol and a dearth of nucleation mode aerosol, since nucleation of new particles is inhibited by precursor vapours instead condensing onto pre-existing aerosol (discussed in more detail in below).

CO and rBC concentrations were strongly correlated ($r^2$ = 0.83) and reached 0.626 ppmv and 1,490 ng kg$^{-1}$, respectively (Figure 7), which are higher than standard summertime and even springtime haze concentrations. The modified combustion efficiency (MCE), defined as $\Delta CO_2/(\Delta CO_2 + \Delta CO)$ where $\Delta CO_2$ and $\Delta CO$ indicate the increase from background $CO_2$ and CO concentrations, respectively, was calculated for data in which fires impacted the measurements (Liu et al., 2014).



Background $CO_2$ and CO concentrations of 383 and 0.054 ppmv, respectively, were defined from the current measurements. These values were derived from correlations of rBC mass versus $CO_2$ and CO, and finding the minimum value of $CO_2$ and CO on the rBC axis. Measurements impacted by fires were characterized as rBC mass and CO $\geq$ 20 ng kg$^{-1}$ and $\geq$ 0.1 ppmv, respectively. MCE values during measurements impacted by fires were close to 1, indicating active flaming (i.e., "fresher" fires) versus smoldering, particularly during flights 13, 17, and 18. CO is a poor tracer for oil extraction since it originates from combustion, thus aside from the operational vehicles in Prudhoe Bay, we would expect the boreal fires to most strongly influence CO during the campaign (Crutzen et al., 1979; Andreae and Merlet, 2001). Background CO concentrations have been measured at 0.120 ppmv using summertime surface measurements at Utqiaġvik and up to 0.4 ppmv in ARCPAC airborne measurements of springtime long-range-transported biomass burning plumes (Liang et al., 2004; Brock et al., 2011). Brock and colleagues reported springtime rBC mass concentrations of up to 1,000 ng m$^{-3}$ using the SP2 instrument also used in the current study. Our observations are parallel to previous summertime observations from regional boreal fires in that they produce substantial quantities of aerosol (Stohl et al., 2006; Eck et al., 2009), which is likely due to the proximity of the measurements to the source.

Notably, anomalously high rBC and CO concentrations were measured during a few flights (F13, F17, and F18). Almost all measurements from the 30 Jun flight were considerably high—including CO (0.626 ppmv), rBC (1490 ng kg$^{-1}$), aerosol number concentrations (20,596 cm$^{-3}$ for CPC 3010, 11,021 cm$^{-3}$ for PCASP, and 30 cm$^{-3}$ for OPC), absorption (61.1 Mm$^{-1}$), and scattering (978.1 Mm$^{-1}$)—as compared to other flights impacted by the fires (Figure 8). AOD was relatively high ($>$ 0.1) and a fire was detected by MODIS close to Toolik Lake (Figure 5) on 30 Jun, which is likely why the measurements were highest when spiralling over the waypoint and then decreased as the aircraft flew low to/from adjacent waypoints. However, the plume on 30 Jun also reached higher altitudes above Oliktok Point (as supported by the MODIS satellite observations), indicating the biomass burning plume ascended as it propagated northward. The only exception to the considerably high nature of the aerosol measurements is nucleation mode aerosol (i.e., CPC$_{diff}$), which was only slightly elevated in concentration (maximum of 2,500 cm$^{-3}$ as compared to a maximum of 101,940 cm$^{-3}$ for the same-sized particles from Prudhoe Bay) and was not elevated over Oliktok Point, demonstrating the short lifetimes of these small-sized particles (via growth into the accumulation mode) in densely-populated biomass burning plumes.

The large quantity of aerosol observed from the lowest to highest altitudes flown by the aircraft closest to the Brooks Range indicate the thickness of the biomass burning aerosol layer. These particles can have implications both on the local energy budget and on cloud formation. We observed how aerosols from the 30 Jun fire event were highly scattering and absorbing relative to the rest of the region (Figure 8e and f, respectively). Both organic and inorganic components of aerosols from wildfires can be highly hygroscopic and serve as efficient CCN (Novakov and Corrigan, 1996; Petters et al., 2009a; Engelhart et al., 2012), while mineral dust, carbonaceous, and biological aerosols from wildfires have been shown to increase atmospheric INP concentrations (Petters et al., 2009b; McCluskey et al., 2014). Additionally, ejection of such a large quantity of aerosol and trace gases into the atmosphere can affect air quality on the North Slope and to the Arctic beyond over the course of a couple of weeks (Stohl et al., 2013).



### 3.3 Relative contributions from regional and long-range-transported sources of aerosol to North Slope

Weaker poleward advection and strong aerosol removal via wet deposition enable the Arctic to be subject to less transport from midlatitude sources during the summer as compared to the spring Arctic haze season (Polissar et al., 2001; Bian et al., 2013). During the summer, aerosol production from local natural sources—from terrestrial and marine microbial processes

and mechanical generation of sea salt—is dominant at the ground and aloft (Gregory et al., 1992; Quinn et al., 2002; Leaitch et al., 2013; Leaitch et al., 2016; Burkart et al., 2017). Although the presence of pollutants from mid-latitudes is typically lower during the summer in the Arctic (Raatz and Shaw, 1984), we observed occasional episodic increases in accumulation and coarse mode aerosol measured by the PCASP and OPC at higher altitudes (Figure 9), without the presence of Prudhoe Bay or Alaskan fire tracers.  These layers were deficient in CO, rBC, and/or 3 – 10 nm particles, and were present during

flights where biomass burning was not detected as a dominant source. Thus, we assume these events were not a result of local or regional emissions that dominated the North Slope aerosol during ACME-V. These diagnosed long-range transport events were only observed during flights 1, 9, 10, and 11, thus supporting the idea that poleward advection is less frequent and/or wet removal processes are enhanced in the summer as compared to the Arctic haze season.

Recent studies have alluded to the possibility that the Arctic summer may not be as pristine as previously thought (Stohl

et al., 2013). Modelled emissions from ARCTAS-A highlight summertime boreal fires and their impact on Arctic pollution, however, these flights targeted local fire plumes and were limited to the Canadian Arctic. Further, Bian and colleagues (2013) state that the ARCTAS-A measurements, "cannot provide a comprehensive and representative picture of Arctic pollution in the summer". We evaluated all ACME-V data and classified each flight as impacted by fires, Prudhoe Bay (PHB), long range transport (LRT), and/or some combination of these (Figure 10a). All flights contained at least a small

segment that was classified as background (i.e., not classified as fire, Prudhoe Bay, or long-range transport), but flights where only background conditions were observed are labelled as background. Due to the aircraft being based in Deadhorse, emissions from Prudhoe Bay impacted nearly every flight (31 flights; remaining 7 flights were flagged for clouds near Prudhoe Bay, thus those data were eliminated from the analysis), while regional fires impacted 22 flights, and long-range transport impacted only 4 flights. For a more statistical representation of the sources, we classified each 1-second data point

as influenced from fires at all flight locations, fires from the lowest latitude flown to 69°N (i.e., a subset of all fires), Prudhoe Bay emissions at all flight locations, emissions strictly near Prudhoe Bay, emissions near Prudhoe Bay in the boundary layer, long-range transport, background, and pristine conditions (Figure 10b; see Table 2 for classification definitions). Background conditions were simply determined by subtracting the percentage of all sources (PHB + fires + LRT + pristine) combined from 100, while pristine conditions represent data left after all aerosol parameter thresholds were applied—aside

from $CPC_{diff}$ concentrations to ensure inclusion of marine aerosol production (Quinn et al., 2002). Background may include aerosol from Prudhoe Bay or the boreal fires after significant atmospheric residence time, but we cannot distinguish these from local natural aerosol emission or production that is traditionally observed with the measurements obtained. This plot demonstrates the episodic behaviour of the fires and the localized behaviour of Prudhoe Bay emissions. However, what we



are calling "pristine" conditions had the lowest occurrence overall (only 5%), which contrasts with previous North Slope summertime aerosol studies. It is important to note that these data are dependent on the location and height of the aircraft, thus may be biased. However, it provides a general overview of the sources of aerosols in the context of the flight locations, but may not be representative of the entire North Slope at all times.

## 4 Summary

Results from the 2015 airborne ARM ACME-V field campaign demonstrate that the summer in the Alaskan Arctic is not necessarily characterized by clean conditions. The pristine nature of the atmosphere is dependent on the influence from episodic wildfires, localized oil extraction activities, and to a lesser extent, long-range-transport. Probably the most notable observation was that Prudhoe Bay is a persistent but localized source of black carbon and especially nucleated aerosol, supporting previous findings at Utqiaġvik from Kolesar and colleagues (2017) and Gunsch and colleagues (2017), but demonstrating the larger impact of nucleated aerosol in the vicinity surrounding Prudhoe Bay. Such elevated aerosol levels have been shown to alter the microphysics of clouds in this region, impacting their radiative forcing on the surface (Maahn et al., 2017). Most previous measurements along the North Slope have been conducted at the ground at a single location, thus thwarting the evaluation of the spatiotemporal heterogeneity of aerosol in the entire Alaskan Arctic. Although our results reveal that these sources are not significant on a larger scale (i.e., the entire North Slope), they yield valuable information on local and regional Arctic pollution sources, which produce substantial quantities of aerosols that may be transported downstream and beyond. Further, although our observations are limited to the location and dynamical conditions of the North Slope, they can serve as a proxy for other parts of the Arctic subject to oil exploration or boreal fires. With both fire activity and oil exploration projected to increase in a warming climate, these sources will likely continue to make significant contributions to the aerosol population of the Arctic atmosphere. The particles emitted from these sources can impact atmospheric radiative transfer through modulation of cloud microphysics and direct radiative forcing. To fully understand the impacts of these particles and their relative frequency of occurrence, additional observational, modelling, and theoretical studies are required.

## Author contributions

J. Creamean analysed and interpreted ARM ACME-V aerosol data, MODIS AOD and thermal anomaly data, and ran HYSPLIT simulations and wrote the manuscript. M. Maahn participated in ACME-V aerosol and HYSPLIT data analyses, and found oil well data. G. de Boer and A. McComiskey helped with general interpretation of the ACME-V data. A. Sedlacek and Y. Feng helped with MCE calculations and interpretation of the SP2 measurements. All co-authors contributed to the writing of or provided comments for the manuscript.



## Acknowledgements

This analysis was supported by funding from the US Department of Energy's Office of Science under the Atmospheric System Research (ASR) program (grant DE-SC0013306). We would like to acknowledge those involved with execution of the ARM ACME-V field campaign, including Sebastien Biraud, Margaret Torn, Duli Chand, Connor Flynn, John Hubbe,

Fan Mei, Mikhail Pekour, Stephen Springston, and Jason Tomlinson. Additionally, we gratefully acknowledge the NOAA Air Resources Laboratory (ARL) for the provision of the HYSPLIT transport and dispersion model and/or READY website (http://www.ready.noaa.gov) used in this publication. Lastly, we acknowledge the use of data and imagery from Land, Atmosphere Near real-time Capability for EOS Land, Atmosphere Near real-time Capability for EOS (LANCE FIRMS) operated by the NASA/GSFC/Earth Science Data and Information System (ESDIS) with funding provided by NASA/HQ.

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



**Table 1. Flight identification numbers, start and end times in both UTC, flight duration, and waypoints flown over of each G1 research flight during ARM-ACME-V. Waypoints O, U, A, I, and T represent Oliktok Point, Utqiaġvik, Atqasuk, Ivotuk, and Toolik Lake, respectively.**

| Flight ID | Start(UTC) | End (UTC) | Duration (hh:mm:ss) | Waypoints |
|---|---|---|---|---|
| F01 | 06/04 22:31:21 | 06/05 01:59:40 | 03:28:19 | O, U |
| F02 | 06/07 20:08:51 | 06/08 01:16:28 | 05:07:37 | O, U, A, I, T |
| F03 | 06/08 19:56:37 | 06/09 00:52:47 | 04:56:10 | O, U, A, I, T |
| F04 | 06/10 18:39:18 | 06/10 20:30:59 | 01:51:41 | O |
| F05 | 06/12 21:58:20 | 06/13 00:11:26 | 02:13:06 | O |
| F06 | 06/13 18:57:18 | 06/13 23:57:32 | 05:00:14 | O, U, A, I, T |
| F07 | 06/15 21:57:26 | 06/16 00:35:49 | 02:38:23 | O, U |
| F08 | 06/17 18:59:36 | 06/18 00:11:03 | 05:11:27 | O, U, A, I, T |
| F09 | 06/20 19:00:37 | 06/21 00:04:12 | 05:03:35 | O, U, A, I, T |
| F10 | 06/22 23:17:43 | 06/23 01:19:26 | 02:01:43 | O |
| F11 | 06/23 19:14:17 | 06/24 00:19:21 | 05:05:04 | O, U, A, I, T |
| F12 | 06/27 21:06:18 | 06/27 23:12:53 | 02:06:35 | O, U |
| F13 | 06/30 18:59:46 | 06/30 22:30:34 | 03:30:48 | O, I, T |
| F14 | 07/02 19:34:03 | 07/02 23:31:36 | 03:57:33 | O, U, T |
| F15 | 07/05 18:57:14 | 07/06 00:10:57 | 05:13:43 | O, U, A, I, T |
| F16 | 07/11 20:27:14 | 07/12 00:51:07 | 04:23:53 | O, U, A, T |
| F17 | 07/14 19:01:15 | 07/14 21:18:25 | 02:17:10 | O, T |
| F18 | 07/16 19:58:43 | 07/17 00:38:47 | 04:40:04 | O, U, A, I, T |
| F19 | 07/18 19:54:33 | 07/19 00:19:00 | 04:24:27 | O, U, A, I, T |
| F20 | 07/21 19:24:37 | 07/22 00:30:44 | 05:06:07 | O, U, A, I, T |
| F21 | 07/22 19:29:54 | 07/23 00:14:20 | 04:44:26 | O, U, A, I, T |
| F22 | 07/27 21:34:03 | 07/28 00:07:11 | 02:33:08 | O, U |
| F23 | 07/30 21:18:11 | 07/31 01:03:48 | 03:45:37 | O, U, A, T |
| F24 | 08/02 18:10:47 | 08/02 21:40:37 | 03:29:50 | O, U, T |
| F25 | 08/06 19:01:12 | 08/06 23:34:33 | 04:33:21 | O, U, A, I, T |
| F26 | 08/07 18:37:02 | 08/07 19:46:35 | 01:09:33 | O |
| F27 | 08/08 19:42:22 | 08/08 21:52:35 | 02:10:13 | O, U |
| F28 | 08/14 18:49:22 | 08/14 22:17:13 | 03:27:51 | O, U |
| F29 | 08/16 19:54:12 | 08/16 23:53:40 | 03:59:28 | O |
| F30 | 08/20 20:47:02 | 08/21 00:09:46 | 03:22:44 | O, U, A, I, T |
| F31 | 08/25 19:18:13 | 08/25 23:52:06 | 04:33:53 | O, U, A, I, T |
| F32 | 08/27 21:29:46 | 08/28 01:37:41 | 04:07:55 | O, U, A, I, T |
| F33 | 08/28 22:30:14 | 08/29 00:51:35 | 02:21:21 | O |
| F34 | 08/30 21:49:18 | 08/30 23:39:16 | 01:49:58 | O |
| F35 | 09/02 19:00:45 | 09/02 23:27:41 | 04:26:56 | O, A, I, T |
| F36 | 09/04 21:24:39 | 09/05 00:47:02 | 03:22:23 | I, T |
| F37 | 09/07 18:28:10 | 09/07 21:23:57 | 02:55:47 | O, A |
| F38 | 09/09 18:29:19 | 09/09 22:18:46 | 03:49:27 | O, U, A, I |





**Table 2. Classification parameters and thresholds for characterization of 1-second data as being impacted by one (or more) of the sources shown in Figure 10b. "PHB", "BL", and "LRT" represent Prudhoe Bay, boundary layer, and long-range transport, respectively.**

| Source | Classification parameter and threshold |
|---|---|
| All fires | 1) $rBC \geq 20$ ng kg$^{-1}$* & <br> 2) $CO \geq 0.1$ ppmv |
| Fires south | 1) $rBC \geq 20$ ng kg$^{-1}$ & <br> 2) $CO \geq 0.1$ ppmv & <br> 3) Latitude $\leq 69°N$ |
| All PHB | 1) $CPC_{diff} \geq 100$ cm$^{-3}$ |
| Near PHB | 1) $CPC_{diff} \geq 100$ cm$^{-3}$ & <br> 2) Distance from Deadhorse < 50 km |
| BL PHB | 1) $CPC_{diff} \geq 100$ cm$^{-3}$ & <br> 2) Distance from Deadhorse < 50 km & <br> 3) Altitude $\leq 500$ m AMSL |
| LRT | 1) $PCASP \geq 600$ cm$^{-3}$** & <br> 2) $CO < 0.1$ ppmv |
| Pristine | 1) $rBC < 20$ ng kg$^{-1}$ & <br> 2) $CO < 0.1$ ppmv & <br> 3) $PCASP < 600$ cm$^{-3}$ & <br> 4) Distance from Deadhorse > 50 km |

5    *Maahn et al. (2017)
     **Based on threshold value selected from Figure 9





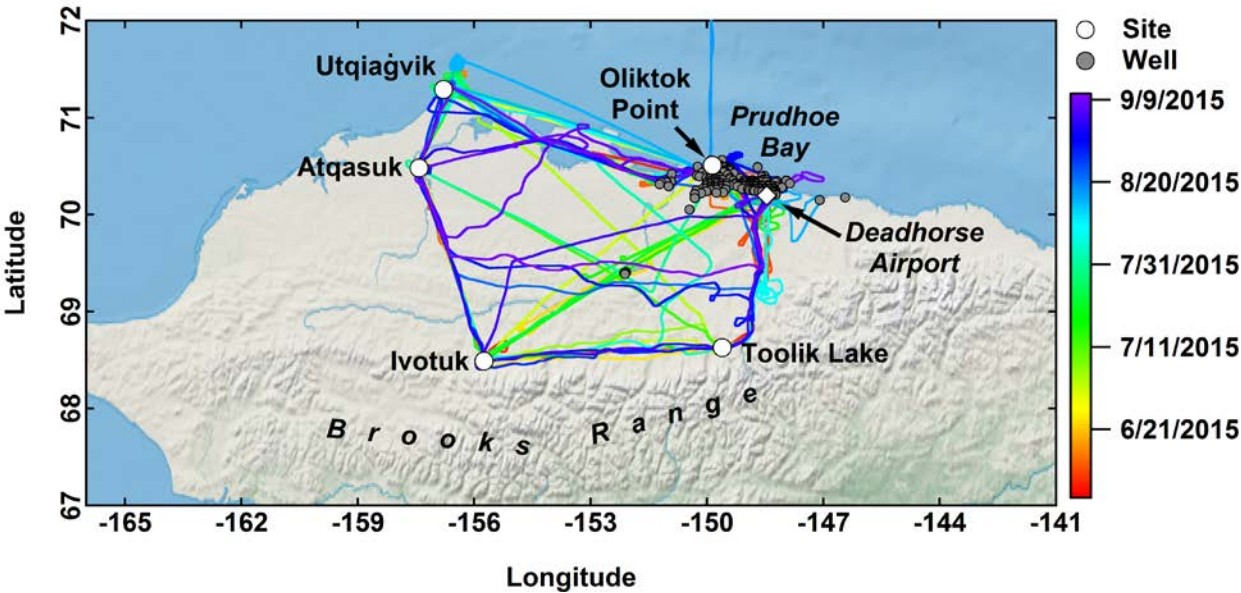

**Figure 1. Map of Alaska including flight tracks from the ARM ACME-V field campaign (2015) coloured by date, sites where the G-1 aircraft spiralled over (Oliktok Point, Utqiaġvik, Atqasuk, Ivotuk, and Toolik Lake), locations of actively deployed oil wells (data obtained from http://doa.alaska.gov/ogc/publicdb.html), location of Deadhorse Airport, and approximate areas of the Brooks Mountain Range and Prudhoe Bay.**





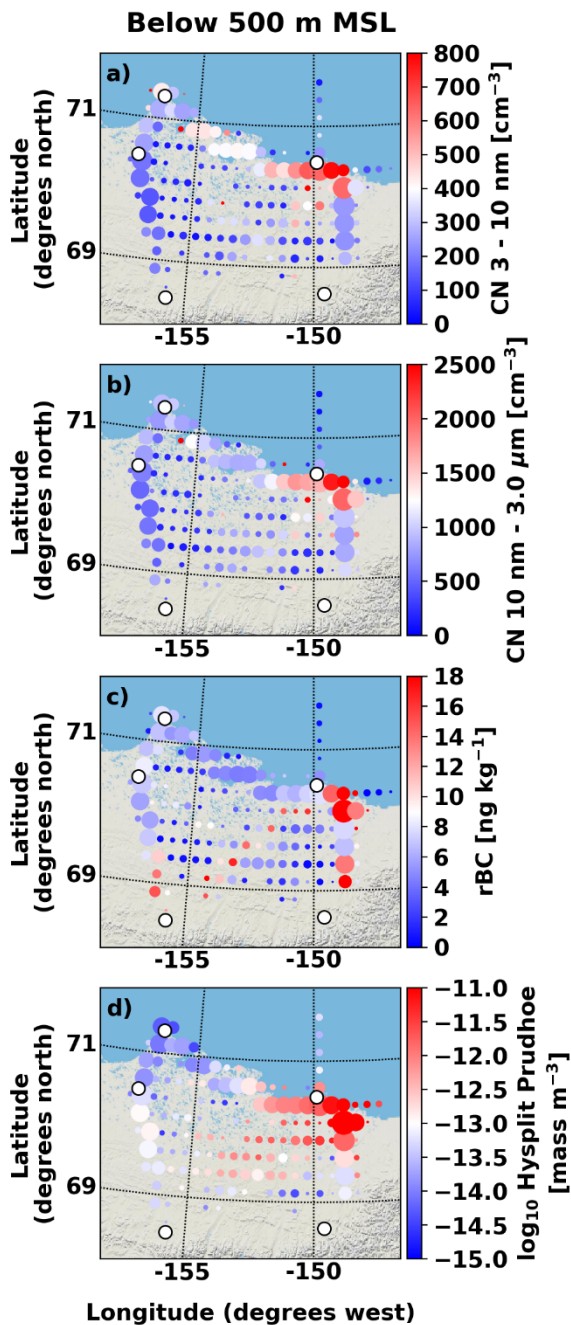

**Figure 2. Maps of column average values for a) CN with $D_p$ = 3 – 10 nm (CPC$_{diff}$), b) CN with $D_p$ = 10 nm – 3 μm (CPC 3010), c) rBC mass (SP2), and d) HYSPLIT aerosol mass concentrations with Prudhoe Bay as the simulation start point for the entire ACME-V campaign from 20 to 500 m AMSL. The size of the marker equates to the number of measurements at each 0.25° latitude x 0.50° longitude grid point. The five white markers show each of the site waypoints.**

**Figure 3. Vertical profiles of particle number concentrations for CN with $D_p$ = 3 – 10 nm (CPC$_{diff}$) and CN 10 nm – 3 μm (CPC 3010) coloured by distance from Prudhoe Bay (a and b, respectively) and by flight number (c and d, respectively). All data shown are 1-second.**



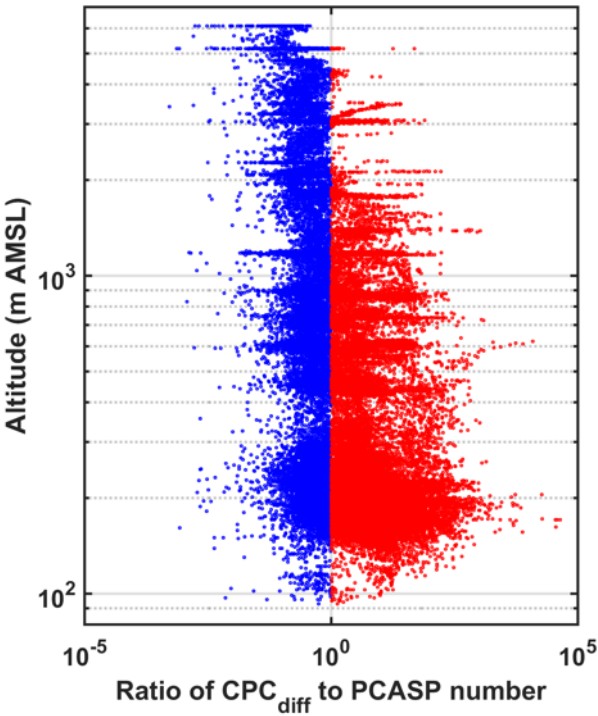

**Figure 4. Ratio of the number concentrations of particles 3 – 10 nm in diameter CPC$_{diff}$ to PCASP number concentrations within 50 km of Deadhorse Airport. Blue indicates a ratio < 1 (i.e., PCASP > CPC$_{diff}$) and red indicates a ratio > 1 (i.e., PCASP < CPC$_{diff}$).**





**Figure 5. Maps of AOD and fires (i.e., thermal anomalies; orange diamonds) detected by MODIS for 11 Jun – 12 Aug time period, showing the transition from few fires to the highest density of fires and back. Flight tracks (green lines) and site locations (white circles) are also shown during each corresponding time period. The black markers in the top**
5 **left panel signify HYSPLIT fire start point locations. The fire location in the circle denotes the fire closest to Toolik Lake on 30 Jun (top right panel).**





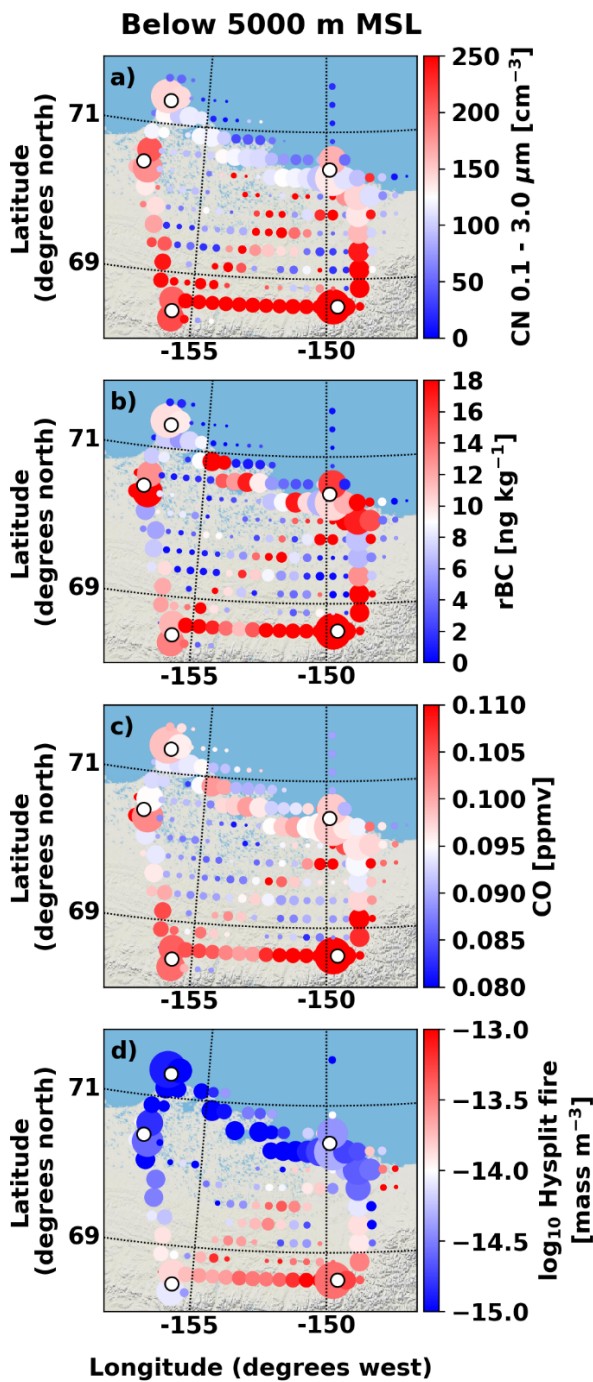

**Figure 6. Same as Figure 2, but column averaged from 20 to 5,000 m AMSL and for a) CN with $D_p = 0.1 – 3$ μm, b) rBC mass, c) CO, and d) HYSPLIT aerosol mass concentrations with the five fire locations as the simulation start points.**





**Figure 7. a) Vertical profile of CO concentrations, b) correlation between CO and rBC mass, and c) correlation between CO and absorption coefficient at 550 nm for each flight during the entire ACME-V campaign. Also shown is the calculated modified combustion efficiency (MCE) for data points classified as being impacted by the fires. All 1-second data are coloured by flight number.**





**Figure 8. 4D profiles of a) CO, b) rBC mass, c) CN with $D_p = 0.1 – 3$ µm, d) CN with $D_p = 0.8 – 15$ µm, e) absorption coefficient, f) scattering coefficient, g) CN with $D_p = 3 – 10$ nm, and h) CN with $D_p = 10$ nm $– 3$ µm from F13 on 30 June 2015. The left, bottom, and right axes are altitude, longitude, and latitude, respectively. Data are 1-second.**



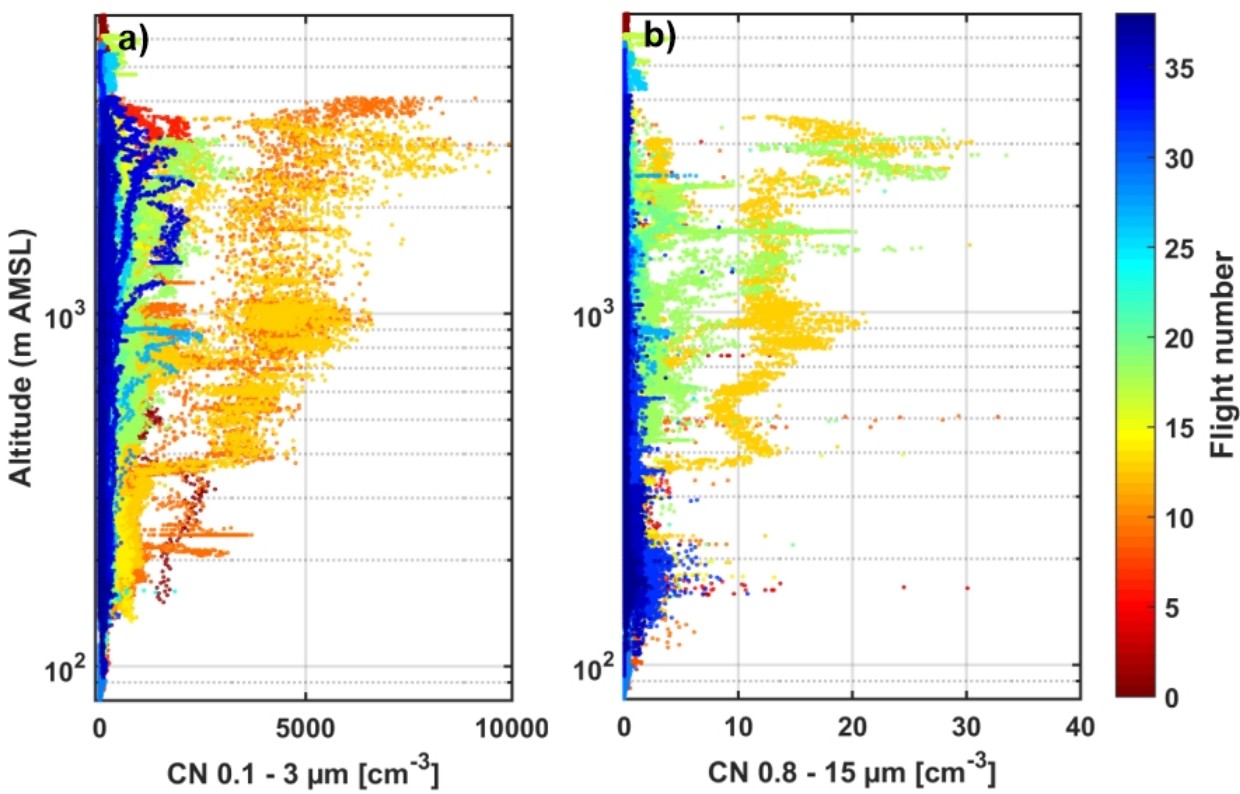

**Figure 9. Vertical profiles of particle number concentrations measured from the a) CN with $D_p = 0.1 - 3$ μm and b) CN with $D_p = 0.8 - 15$ μm colored by flight number. Data shown are 1-second.**



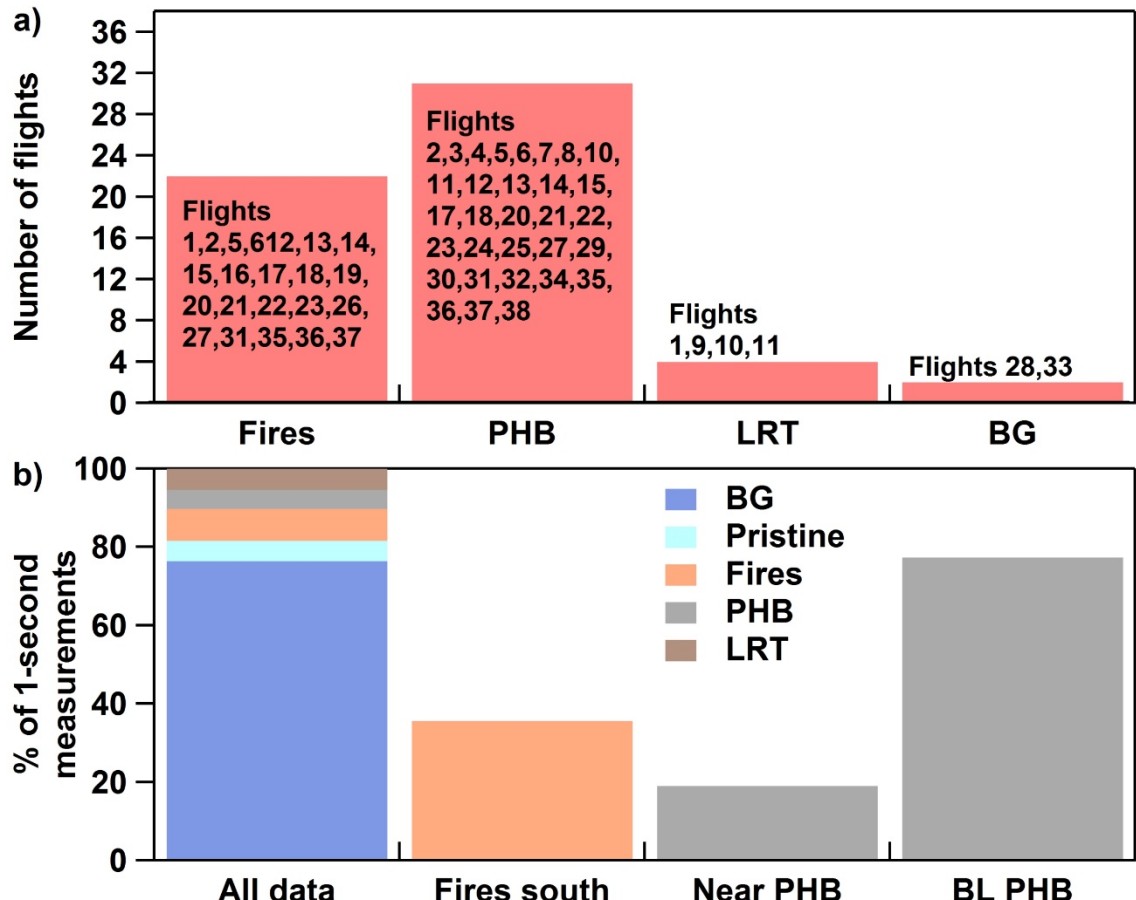

**Figure 10. a) The number of flights that were classified as impacted by aerosols from the central Alaskan fires, Prudhoe Bay (PHB), long-range transport (LRT), background (BG; with no influences from fires, PHB, or LRT), and pristine conditions (based on parameter thresholds in Table 2). b) Percentage of 1-second measurements that were classified as impacted by fires, PHB, LRT, BG, and pristine conditions. The percentage of 1-second fire data points (i.e., from the orange portion in the first bar in b), but for fires south (of 69°N) are shown (second bar). The percentage of 1-second PHB data points (i.e., from the grey portion in the first bar in b), but for data near PHB (within a 50 km radius; third bar) and near PHB + in the boundary layer (BL; fourth bar) near PHB (within a 50 km radius and < 500 m AMSL) are also shown.**