# Peer review of "The influence of local oil exploration and regional wildfires on summer 2015 aerosol over the North Slope of Alaska"

_Atmospheric Chemistry and Physics, 2017_

## Referee Comment (RC1) · Anonymous Referee #1 · 8 Aug 2017

**Comments on "The influence of local oil exploration, regional wildfires, and long range transport on summer 2015 aerosol over the North Slope of Alaska" by Creamean et al.**

This study analysed the airborne observations during ACME-V campaign along the North Slope of Alaska in the summer of 2015 and found that summertime Alaskan Arctic was not pristine as suggested by previous evidence, but was with higher aerosol loading and trace gas concentrations than measurements even in Arctic haze. Local oil extraction activities, central Alaskan wildfires, and to a lesser extent, long-range transport enhanced the aerosol and trace gas concentrations in Alaskan Arctic during summertime. Quantifying aerosol loading and sources in the Arctic is challenging. The aircraft observations presented in this study is therefore an important contribution to the field, but the analysis and writing quality of this manuscript is really poor. I recommend publication in ACP after major revisions and substantial improvements.

**Major comments**:

1. The analysis of the data was a little superficial and I suggest the authors dig deeper. For example, in Figs. 3, 7 and 9, the data were color coded by flight numbers, which does not provide any valuable information. They already classified the flights into several air mass types as shown in Table 2 and Fig. 10. I think analysis based on different air mass types would provide more information than the current flight numbers used in the manuscript. In addition, Figs. 2, 3 and 4 discussing the impacts of oil extraction was based on data during the whole campaign. I suggest select the period during which these sources dominate would be better to illustrate their contributions.

2. The manuscript was poorly organized, making it really hard to follow. For example, in Sect. 3.1, figures were discussed back and forth. Fig. 2 c and d were discussed after Fig. 4. In the same section, the idea that 'high concentration of small particles are restricted within 50 km of Deadhorse' has been discussed several times (P6, L21–26, P7, L14–16, and P7, L24–25). In Sect. 3.2, discussions of different species were also jumped back and forth. For instance, aerosols were discussed in P8, L15–19, P8, L25–29 and P9, L21–25. Background concentrations of CO and enhanced CO were discussed back and forth in P9, L1–14. In Sect. 3.3, the second paragraph discussing air mass types along the flights does not belong to this section, which is supposed to discuss the contribution from long range transport. Long range transport deserves more analysis.

3. Another problem of the paper is the sloppy style of writing and the use of the English language. For instance, the tense was wrong in numerous places. To name a few, P1, L 22–44, P3, L16, and P3, L30. The references were not always written in the correct format. '… and colleagues (year)' should be '… et al. (year)'. The acronyms were not properly used (e.g. 'rBC' and 'black carbon', 'CO' and 'carbon monoxide'

were used back and forth; AMSL and MSL were not spelled out when they were used for the first time; ARM and AOD were spelled out twice). A lot of 'and/or' were used. Please double check and delete 'and' or 'or'. I also list a few other problems in the 'Minor comments' section, but all these I've pointed out are only a few of the language problems in the manuscript. I suggest a much more careful checking of the manuscript and a substantial improvement of the language.

4. In section 3.2, please compare the fire activity in summer 2015 with climatology to illustrate how representative the summer is.

5. Axis labels of Fig. 7b are wrong.

**Minor comments**:

1. P2, L22: 'to discover' is inappropriate here. Revise please.

2. P2, L25: 'during their Aug–Sep 2015 study' -> 'during Aug–Sep, 2015'

3. P2, L29: 'exists' -> 'locates'

4. P2, L31: revise 'provides the ability to …'

5. P2, L33: 'long-range transported aerosol from lower latitudes' -> 'long-range transport from lower latitudes'

6. P3, L1: 'insight' -> 'insights'

7. P3, L2–3: please provide proper references

8. P4, L23: $CO_2$ were also discussed.

9. P5, L6: particles -> particle

10. P5, L23: 'landing the Deadhourse' -> 'landing in the Deadhourse''

11. P5, L30–33: please show these locations in related figures.

12. P6, L17–18: please clarify which data were used.

13. P6, L19: '&'-> 'and'

14. P6, 31: those vapours does not nucleate, the secondary products are. Please clarify.

15. P8, L31: Please provide concentration values in standard summertime and springtime.

16. P9, L4–5: please clarify whether it is active flaming or smoldering.

17. P10, L9: what are the tracers?

18. P19: Figure 2. [mass m$^{-3}$] is not a unit

19. P22: Figure 5. Move the colour bar to the bottom of the figure.

---

## Referee Comment (RC2) · Anonymous Referee #2 · 12 Aug 2017

This paper presents measurements of aerosols made during the U.S. Department of Energy Atmospheric Radiation Measurement (ARM) program's Fifth Airborne Carbon Measurements (ACME-V) campaign along the North Slope of Alaska during the summer of 2015. The paper focuses on how local oil extraction activities long-range transport influence aerosols and trace gases in the North Slope of Alaska. The authors should try to go beyond presenting the measurements and use the data in a clearer way to demonstrate the scientific conclusions that can be made using the data. This paper is within the scope of ACP and should be published following after the authors address the following comments:
* * *
Interactive
comment

1. I agree with reviewer #1 that the authors should rethink how to present the data in a less superficial way in addition to showing the data as a function of the flight number. I don't object to showing these figures (Figs. 3, 7 and 9) as long as the data is shown again in a more synthetic way later in the paper, allowing the authors to draw more clear conclusions from the measurements.

2. The choice for the classification parameters and thresholds values in Table 2 should be more clearly justified. Have these been chosen using the Hysplit analysis?

3. The way that Hysplit has been run should be more clearly described and justified. Even though the authors reference another paper for the description of the Hysplit runs, there is not enough information to fully understand how Hysplit was run. I assume this was run in backward mode from the measurement locations, but this is not clear. The reason for the choice of the five locations in the active fire region is also not clear. I also cannot fully understand Figure 2d and Figure 6d.

4. The reason for showing the data as column averaged values in Figures 2 and 6 needs to be justified. Don't we lose information by showing the data in this way? The main advantage of using aircraft data is that we know where the aerosol layers are vertically. The information we can learn from the altitude of the aerosol layers should be a clearer part of this analysis.

5. The MODIS detected fire hotspots should be shown on Figure 5 relative to the fire size or fire radiative power, such that more active fires can be identified vs. less active fires.

6. The influence of oil exploration is not clear to me. Is the location of oil exploration activities known? The discussion of oil exploration influenced air that was sampled should be clarified. The discussion of long range transport also needs to be developed, as noted by reviewer #1.

7. The authors should review the manuscript writing to clean up the writing style and

typos before resubmission.

---

## Referee Comment (RC3) · Anonymous Referee #3 · 15 Aug 2017

Comments on Creamean et al., 2017 –Anonymous referee 3

This study reports new aerosol observations from the summertime ARM ACME-V field campaign on the North slope of Alaska. Their results indicate that oil exploration activities in Prudoe bay may contribute significantly to local aerosol concentrations. In particular, they observed high concentrations of nucleation mode particles in the area. These observations are novel and interesting and certainly add to the scientific discourse. However, many of the conclusions presented I feel are not fully justified by the results discussed. Additionally, the paper is difficult to read in several sections and requires clarification on several points. Please see below for specific instances.

[Figure]

Major comments

Page 2 lines 3-4 This entire sentence is very confusing, what do the authors mean by 'beyond greenhouse gases' and which 'climate feedback' are they referring to? Please refer to the specific feedback (I assume ice-albedo) and rewrite the sentence to improve clarity.

Page 2 line 5 'indirectly impact radiation through their role in cloud lifecycle' Aerosol can indirectly result in radiative forcing by increasing cloud lifetime, changing albedo and (in the case of INP) changing cloud phase. Please rewrite the sentence to address generalities (cloud microphysics) or specific processes.

Page 3 line 3 'The Arctic atmosphere can be highly stratified' This statement is included without explanation or more (crucially) reference. Please cite a supporting reference.

Page 3 lines 3-4 'at the height at which cloud modulation by aerosols occurs' This statement is very vague. To what height are you referring? Cloud base in the Arctic can be extremely low.

Page 3 lines 4-5 'have focused in evaluating Alaskan Arctic aerosol sources' Why would focusing on aerosol sources help our understanding of higher altitude aerosol in the Arctic? I assume the authors mean specifically sources of higher altitude aerosol?

Page 3 lines 23-26 This statement seems to be conflating future and present sources of aerosol in the Arctic and while forest fires are indeed an important source the relative importance of local vs transported aerosol is not well understood. Thus, while 'great importance' may be justified for boreal forest fires (during some periods of the year) I would prefer more nuance when discussing local fossil fuel and BB combustions (by which I'm assuming the authors mean domestic wood burning?).

Page 4 line 7 'predominantly decoupled' Why is the summertime Arctic decoupled? And is this also true for the sub-Arctic region that this paper focuses on? If the Arctic is less polluted in the summer (because as your references suggest it is decoupled from

the mid-latitudes) does this not suggest that local sources are unimportant?

Page 6 lines 23-24 'Hotspots of larger particles. . ... were not observed near Prudhoe bay (not shown)' I don't understand why you include this sentence were you expecting to see larger particles?

Page 7 lines 2-10 Here you suggest that predominance of larger particles above the BL is the result of growth during vertical transport and dynamical restriction of nucleation mode particles in the BL. It surely can't be the result of both?

Page 8 lines 4-9 beginning 'in terms of indirect forcing' Is the argument here that accumulation mode aerosol derived from nucleation have an impact on cloud properties or that the nucleation mode particles affect cloud directly? If the latter, please explain why marine aerosol may be more hydroscopic then sulphate. Petter et al., (2007, ACP) suggest similar kappa values for both.

Page 8 line 9-11 beginning 'In general, our results' Given your previous statement that the aerosol appears to have a trivial direct forcing effect and extremely uncertain indirect effect is your final statement really justified?

Page 8 lines 27-29 Why would you expect to observe an 'abundance of coarse and accumulation mode particles' if the wildfires generate secondary organic aerosol?

Page 8 line 31 'Which are higher than standard summertime and even springtime haze concentrations' what is the 'standard' concentration and why have you not provided citations here?

Page 9 lines 11-13 'Our observations are parallel to previous summertime observations from regional boreal fires in that they produce substantial quantities of aerosol. . ..' I'm genuinely unsure what you mean by this sentence although I am confident that it is not your observations producing aerosol. Please clarify.

Page 11 lines 10-11 ',but demonstrating the larger impact of nucleated aerosol in the vicinity surrounding Prudoe bay' Assuming you mean climate impacts I fail to see how

this study demonstrates any impact from these aerosol. You state on page 7 (lines 31-33) that you didn't observe any direct forcing. I also see no evidence (from these observations) of the indirect impact. Please clarify whether the observations reported here do or do not suggest a significant indirect aerosol forcing from emissions in Prudoe bay and provide greater justification for this conclusion.

Page 11 lines 18-21 'With both fire activity and oil exploration projected to increase in a warming climate, these sources will likely continue to make significant contributions. . . .' Previously the authors have stated that emissions in Prudoe bay have a localized impact only, with this in mind can the authors justify so strong a statement on the importance of future oil exploration to Arctic aerosol?

Minor comments

Page 2 line 3 Missing 'the' before climate feedbacks. However, to improve clarity I would suggest replacing 'climate feedback' with 'ice-albedo feedback' (please see first major comment)

Page 2 line 6 Use of the phrase 'hinges' is colloquial I would suggest changing to 'depends'

Page 2 line 7 'inherently depends' is a redundancy please delete inherently

Page 2 line 7 Replace 'atmospheric processing' with age

Page 4 line 9 'important sources of aerosol' You mean important local sources of summertime aerosol?

Page 4 line 11 To improve clarity please replace 'such' with local

Page 6 line 31- Page 7 line 32 Please split your citations to differentiate between those referencing flaring emissions and those referencing nucleation mechanisms.

Page 7 line 2 Please change 'removal of particles' to 'transfer of particles' or equivalent. Removal suggests removal of the particles from the atmosphere.

[Figure]

Page 7 line 13 Please cite Stohl et al., 2013 (https://www.atmos-chem-phys.net/13/8833/2013/) in reference to BC emissions from flaring

Page 7 line 27 Please replace 'loss' with 'transition'. The particles aren't lost there just bigger.

Page 8 line 7 Are you referring to diameter or radius here in reference to CCN?

Page 8 line 16 'evidenced by the elevated AOD originating from central Alaska' The elevated AOD is the result of aerosol originating from central Alaska. I would suggest rewording to 'the elevated AOD originating from central Alaskan wildfires' or equivalent.

Page 8 line 17 'extended until the end of Jul' What was extended until the end of July?

Page 10 lines1-2 I agree with this statement but both of the papers cited are concerned only with the Alaskan Arctic. I would also suggest citing Garrett et al., 2010 http://journals.co-action.net/index.php/tellusb/article/view/16525/0 , Browse et al., 2012 https://www.atmos-chem-phys.net/12/6775/2012/ or Eckhardt et al., 2003 https://www.atmos-chem-phys.net/3/1769/2003/acp-3-1769-2003.html (among others)

---

## Author Comment (AC1) · 17 Oct 2017

*We thank the reviewers for their insightful feedback. We have substantially revised the manuscript as a result. Mainly, we revised and coordinated almost all of the figures and improved upon the writing quality of the text based on the suggestions provided. We also revised the source classifications and redid any calculations resulting from such changes, all of which are reflected in the manuscript.*

**Reviewer 1**

This study analysed the airborne observations during ACME-V campaign along the North Slope of Alaska in the summer of 2015 and found that summertime Alaskan Arctic was not pristine as suggested by previous evidence, but was with higher aerosol loading and trace gas concentrations than measurements even in Arctic haze. Local oil extraction activities, central Alaskan wildfires, and to a lesser extent, longrange transport enhanced the aerosol and trace gas concentrations in Alaskan Arctic during summertime. Quantifying aerosol loading and sources in the Arctic is challenging. The aircraft observations presented in this study is therefore an important contribution to the field, but the analysis and writing quality of this manuscript is really poor. I recommend publication in ACP after major revisions and substantial improvements.

Major comments:
1. The analysis of the data was a little superficial and I suggest the authors dig deeper. For example, in Figs. 3, 7 and 9, the data were color coded by flight numbers, which does not provide any valuable information. They already classified the flights into several air mass types as shown in Table 2 and Fig. 10. I think analysis based on different air mass types would provide more information than the current flight numbers used in the manuscript. In addition, Figs. 2, 3 and 4 discussing the impacts of oil extraction was based on data during the whole campaign. I suggest select the period during which these sources dominate would be better to illustrate their contributions.

*We agree with the reviewer that the data colored by flight number is not useful and redundant to Table 2 and Figure 9 (was Figure 10). We want to note that the analysis in the manuscript is intended as an overview and presentation of the unique dataset to show the influence of the sources in the Prudhoe Bay area. More detailed studies are currently undergoing the planning phase to elucidate aerosol sources in a more specific manner. However, in an effort to conduct a deeper analysis based on air mass types, we tied in the source classifications more thoughtfully throughout the discussion and respective figures. For clarity and consistency, we revised most of the figures to show all data and those data classified by the air mass types. Much of the text for section 3.1 and some of the text in section 3.2 was updated to reflect the source-specific analysis in the figures. Specifically regarding the figures, we:*

- *Revised the color scheme in Figure 9 (was Figure 10) for each source; now all source colors are consistent throughout all of the figures.*
- *Changed the color scales in Figures 2 and 5 (was Figure 6) to reflect the approximate source colors.*
- *Removed the panels colored by flight number entirely from Figure 3 and colored the data impacted by Prudhoe Bay in blue (all other data in grey).*
- *Combined Figures 3 and 4 into the new Figure 3.*
- *Changed the color scale in Figures 4 and 7 (were Figures 5 and 8) to match fires source color.*
- *Revised Figures 6 and 8 (were Figures 7 and 9) show the vertical profiles and/or correlations for all data and those data impacted by fires and all sources, respectively.*
- *Kept the spatial averaged maps to show, qualitatively, the spatial variability in the parameters and to demonstrate the locations impacted by Prudhoe Bay and the fires. However, the color scales now reflect the assumed sources.*

*With regard to the last point (restricting the oil extraction analysis to the periods during which those sources dominate), it is important to note that this signal is continuous and was encountered by almost all flights as they traversed the Prudhoe Bay area. Therefore, we felt that it was appropriate to highlight the specific signal of these emissions in contrast to the background signal encountered during much of the rest of the flights at altitudes below 500 m.*

2. The manuscript was poorly organized, making it really hard to follow. For example, in Sect. 3.1, figures were discussed back and forth. Fig. 2 c and d were discussed after Fig. 4. In the same section, the idea that 'high concentration of small particles are restricted within 50 km of Deadhorse' has been discussed several times (P6,

L21–26, P7, L14–16, and P7, L24–25). In Sect. 3.2, discussions of different species were also jumped back and forth. For instance, aerosols were discussed in P8, L15–19, P8, L25–29 and P9, L21–25. Background concentrations of CO and enhanced CO were discussed back and forth in P9, L1–14. In Sect. 3.3, the second paragraph discussing air mass types along the flights does not belong to this section, which is supposed to discuss the contribution from long range transport. Long range transport deserves more analysis.

*We went through and reorganized to ensure the figures are discussed in an orderly manner and prevent redundancy in ideas presented. We also added a paragraph describing the classifications of sources in more detail, which helps elucidate the long-range transport analysis. However, we disagree that the second paragraph in section 3.3 does not belong. The focus of the paper is on the abundant local and regional sources (which may be increasingly important in a dynamic Arctic environment), while long-range transport is secondary. Our study and previous studies have indicated that this is not an important source in the summer as compared to the winter/spring. The purpose of this section is to discuss the contributions from all sources compared to one another. A more detailed analysis of long-range transport would require extensive air mass trajectory analysis in addition to other remote sensing or modelling techniques to accurately evaluate long-range sources; this is outside the scope of our manuscript. To reflect the secondary importance of long-range transport, we removed 'long-range transport' from the title.*

*For the figures, we revised so that they are in order when first presented, however, we do refer back to certain figures when discussing different parameters. For Prudhoe Bay, we organized the discussion such that we focus on each measurement parameter (i.e., nucleation mode aerosol and rBC) at a time, which is why we go back and forth between Figures 2 and 3. For the fires, we discussed rBC and CO back and forth because they are related, correlate strongly, and thus both used as tracers for the fires. For both sections, we now show and discuss HYSPLIT first to qualitatively provide spatial evidence of the sources, then discuss how the measurements support the source modeling.*

3. Another problem of the paper is the sloppy style of writing and the use of the English language. For instance, the tense was wrong in numerous places. To name a few, P1, L 22–44, P3, L16, and P3, L30. The references were not always written in the correct format. '… and colleagues (year)' should be '… et al. (year)'. The acronyms were not properly used (e.g. 'rBC' and 'black carbon', 'CO' and 'carbon monoxide' were used back and forth; AMSL and MSL were not spelled out when they were used for the first time; ARM and AOD were spelled out twice). A lot of 'and/or' were used. Please double check and delete 'and' or 'or'. I also list a few other problems in the 'Minor comments' section, but all these I've pointed out are only a few of the language problems in the manuscript. I suggest a much more careful checking of the manuscript and a substantial improvement of the language.

*We cleaned up the writing style throughout the manuscript and made sure we corrected wrong tense usage, citations, and acronym consistency and definitions.*

4. In section 3.2, please compare the fire activity in summer 2015 with climatology to illustrate how representative the summer is.

*We now state that it is the second largest number of acres burned since records began in 1940 based on the findings of Partain Jr., J. L.; Alden, S.; Strader, H.; Bhatt, U. S.; Bieniek, P. A.; Brettschneider, B. R.; Walsh, J. E.; Lader, R. T.; Olsson, P. Q.; Rupp, T. S.; R.L. Thoman, J.; York, A. D.; Ziel, R. H., An Assessment of the Role of Anthropogenic Climate Change in the Alaska Fire Season of 2015. Bulletin of the American Meteorological Society 2016, 97, (12), S14-S18.*

5. Axis labels of Fig. 7b are wrong.

*Fixed.*

Minor comments:
1. P2, L22: 'to discover' is inappropriate here. Revise please.

*Changed to 'to conclude'.*

2. P2, L25: 'during their Aug–Sep 2015 study' -> 'during Aug–Sep, 2015'

*Done.*

3. P2, L29: 'exists' -> 'locates'

*Changed to 'is located'.*

4. P2, L31: revise 'provides the ability to …'

*We removed this sentence and instead combined with the second sentence in the paragraph to, "This site is located in the northwest region of oil extraction activities in Prudhoe Bay, making it an ideal location to determine the potential impacts of emissions from such activities on the relatively pristine Arctic atmosphere."*

5. P2, L33: 'long-range transported aerosol from lower latitudes' -> 'long-range transport from lower latitudes'

*Done.*

6. P3, L1: 'insight' -> 'insights'

*Done.*

7. P3, L2–3: please provide proper references

*Done.*

8. P4, L23: CO2 were also discussed.

*Changed to 'aerosol, CO, and $CO_2$'.*

9. P5, L6: particles -> particle

*This should be particles.*

10. P5, L23: 'landing the Deadhourse' -> 'landing in the Deadhourse''

*Changed to 'landing at the Deadhorse'.*

11. P5, L30–33: please show these locations in related figures.

*These are already shown in Figure 4 (was Figure 5) as indicated in the caption. However, we changed the color of the location markers to make them more evident.*

12. P6, L17–18: please clarify which data were used.

*Clarified that these are thermal anomaly data.*

13. P6, L19: '&'-> 'and'

*Done.*

14. P6, 31: those vapours does not nucleate, the secondary products are. Please clarify.

*Done.*

15. P8, L31: Please provide concentration values in standard summertime and springtime.

*We removed this part of the sentence because the SP2 measures refractory black carbon, and the values in most studies from the North Slope are either equivalent black carbon, or modeled black carbon. Thus, they may not be directly comparable due to possibly slight variations in sampling techniques.*

16. P9, L4–5: please clarify whether it is active flaming or smoldering.

*We already stated that the MCE value indicated active flaming, but changed 'versus' to 'instead of' for clarity.*

17. P10, L9: what are the tracers?

*We added 'CO and rBC' at the end of this sentence.*

18. P19: Figure 2. [mass m-3] is not a unit

*This is the unit defined by the HYSPLIT manual. It is an arbitrary unit. We now describe this in more detail in section 2.4 (was section 2.3).*

19. P22: Figure 5. Move the colour bar to the bottom of the figure.

*Done.*

---

## Author Comment (AC2) · 17 Oct 2017

*We thank the reviewers for their insightful feedback. We have substantially revised the manuscript as a result. Mainly, we revised and coordinated almost all of the figures and improved upon the writing quality of the text based on the suggestions provided. We also revised the source classifications and redid any calculations resulting from such changes, all of which are reflected in the manuscript.*

**Reviewer 2**

This paper presents measurements of aerosols made during the U.S. Department of Energy Atmospheric Radiation Measurement (ARM) program's Fifth Airborne Carbon Measurements (ACME-V) campaign along the North Slope of Alaska during the summer of 2015. The paper focuses on how local oil extraction activities long-range transport influence aerosols and trace gases in the North Slope of Alaska. The authors should try to go beyond presenting the measurements and use the data in a clearer way to demonstrate the scientific conclusions that can be made using the data. This paper is within the scope of ACP and should be published following after the authors address the following comments:

1. I agree with reviewer #1 that the authors should rethink how to present the data in a less superficial way in addition to showing the data as a function of the flight number. I don't object to showing these figures (Figs. 3, 7 and 9) as long as the data is shown again in a more synthetic way later in the paper, allowing the authors to draw more clear conclusions from the measurements.

*Please see response to major comment 1 from reviewer 1.*

2. The choice for the classification parameters and thresholds values in Table 2 should be more clearly justified. Have these been chosen using the Hysplit analysis?

*These are partially based on HYSPLIT analyses, but additionally on thresholds from previous work, and visual assessment of the proximity to known sources and vertical profiles. We added a new paragraph at the end of section 2.4 (was section 2.3) describing Table 2 classifications and how they were derived. In order to follow details on characterization of the fire locations, we moved what was section 2.3 to the end of the methods, after we discuss supporting satellite data.*

3. The way that Hysplit has been run should be more clearly described and justified. Even though the authors reference another paper for the description of the Hysplit runs, there is not enough information to fully understand how Hysplit was run. I assume this was run in backward mode from the measurement locations, but this is not clear. The reason for the choice of the five locations in the active fire region is also not clear. I also cannot fully understand Figure 2d and Figure 6d.

*The new paragraph describing the source classifications now provides justification for the HYSPLIT dispersion analysis. Dispersion simulations are automatically run in forward mode since it simulates emission and transport of particles from a point source. We added more detail in this section describing what information the HYSPLIT dispersion model provides, which also clarifies what is shown in Figures 2a and 5a (were 2d and 6d), in addition to provided more detail on what the model output is. The five locations were chosen based on equal spacing within the highest density of fires determined from the satellite analyses from the entire study time period. This is now stated in the aerosol dispersion modelling section (2.4; was section 2.3).*

4. The reason for showing the data as column averaged values in Figures 2 and 6 needs to be justified. Don't we lose information by showing the data in this way? The main advantage of using aircraft data is that we know where the aerosol layers are vertically. The information we can learn from the altitude of the aerosol layers should be a clearer part of this analysis.

*The purpose of the maps in Figures 2 and 5 (was Figure 6) is to show, qualitatively, the spatial variability in the parameters and to demonstrate the locations impacted by Prudhoe Bay and the fires. Also, the column averaged data in Figure 2 is for altitudes < 500 m AMSL. For Figure 5 (was Figure 6), data are restricted to < 5000 m AMSL to show the vertical extent of the fire impacts. These parameters are shown as vertical profiles of the 1-second measurements in the following figures, thus any information that may be lost in Figures 2 and 5 are shown elsewhere. However, the conclusions discussed for these figures are supported by the vertical analyses.*

5. The MODIS detected fire hotspots should be shown on Figure 5 relative to the fire size or fire radiative power, such that more active fires can be identified vs. less active fires.

*Fire size and radiative power information is not available from the thermal anomaly data we used. Additionally, we used the fires as a qualitative approach to evaluate when and where these sources were present, and used the spatial density of the data to determine where HYSPLIT dispersion simulations should be initiated. Evaluation of fire properties is outside the scope of our manuscript.*

6. The influence of oil exploration is not clear to me. Is the location of oil exploration activities known? The discussion of oil exploration influenced air that was sampled should be clarified. The discussion of long range transport also needs to be developed, as noted by reviewer #1.

*This should be evident now given the additional paragraph describing the source classifications. Additionally, previous work by Gunsch et al. (2017) and Kolesar et al. (2017) clearly demonstrate how oil exploration from Prudhoe Bay is an influence on the North Slope. The locations of the oil activities (i.e., the active oil wells) is provided in Figure 1 and now include access date of the data. Please see response to reviewer 1's comment regarding long-range transport.*

7. The authors should review the manuscript writing to clean up the writing style and typos before resubmission.

*Done.*

---

## Author Comment (AC3) · 17 Oct 2017

*We thank the reviewers for their insightful feedback. We have substantially revised the manuscript as a result. Mainly, we revised and coordinated almost all of the figures and improved upon the writing quality of the text based on the suggestions provided. We also revised the source classifications and redid any calculations resulting from such changes, all of which are reflected in the manuscript.*

**Reviewer 3**

This study reports new aerosol observations from the summertime ARM ACME-V field campaign on the North Slope of Alaska. Their results indicate that oil exploration activities in Prudoe bay may contribute significantly to local aerosol concentrations. In particular, they observed high concentrations of nucleation mode particles in the area. These observations are novel and interesting and certainly add to the scientific discourse. However, many of the conclusions presented I feel are not fully justified by the results discussed. Additionally, the paper is difficult to read in several sections and requires clarification on several points. Please see below for specific instances.

Major comments:
Page 2 lines 3-4 This entire sentence is very confusing, what do the authors mean by 'beyond greenhouse gases' and which 'climate feedback' are they referring to? Please refer to the specific feedback (I assume ice-albedo) and rewrite the sentence to improve clarity.

*We specified that this is the ice-albedo feedback and reworded the sentence to, "In addition to the ice-albedo feedback described above, the principal atmospheric constituents that perturb the surface energy budget are clouds and aerosols."*

Page 2 line 5 'indirectly impact radiation through their role in cloud lifecycle' Aerosol can indirectly result in radiative forcing by increasing cloud lifetime, changing albedo and (in the case of INP) changing cloud phase. Please rewrite the sentence to address generalities (cloud microphysics) or specific processes.

*We rewrote the sentence so that it says 'roles in the modulation of cloud microphysics'.*

Page 3 line 3 'The Arctic atmosphere can be highly stratified' This statement is included without explanation or more (crucially) reference. Please cite a supporting reference.

*We added two key references.*

Page 3 lines 3-4 'at the height at which cloud modulation by aerosols occurs' This statement is very vague. To what height are you referring? Cloud base in the Arctic can be extremely low.

*Good point, Arctic cloud base can indeed be extremely low. We removed 'at the height at which cloud modulation by aerosols occurs' that sentence.*

Page 3 lines 4-5 'have focused in evaluating Alaskan Arctic aerosol sources' Why would focusing on aerosol sources help our understanding of higher altitude aerosol in the Arctic? I assume the authors mean specifically sources of higher altitude aerosol?

*We changed this sentence to read, "Accordingly, numerous airborne campaigns have focused on evaluating sources of mid- to upper-tropospheric aerosol and aerosol-cloud interactions."*

Page 3 lines 23-26 This statement seems to be conflating future and present sources of aerosol in the Arctic and while forest fires are indeed an important source the relative importance of local vs transported aerosol is not well understood. Thus, while 'great importance' may be justified for boreal forest fires (during some periods of the year) I would prefer more nuance when discussing local fossil fuel and BB combustions (by which I'm assuming the authors mean domestic wood burning?).

*We are referring to projected increases in forest fires due to a warming climate. Domestic wood burning should not play a dominant role in the summertime in this region, particularly when compared to the widespread forest fires. However, to make this sentence clearer, we changed to, "In the context of warming temperatures, emissions from oil*

*extraction, added shipping routes due to a reduction in sea ice extent, and wildfires are expected to increase in sub-Arctic boreal regions (Randerson et al., 2006; Gautier et al., 2009; Harsem et al., 2011; Peters et al., 2011; de Groot et al., 2013; Roiger et al., 2015). Thus, regional fossil fuel and biomass burning combustion sources will further contribute to the aerosol population may serve as an increasingly crucial source of aerosol in the future."*

Page 4 line 7 'predominantly decoupled' Why is the summertime Arctic decoupled? And is this also true for the sub-Arctic region that this paper focuses on? If the Arctic is less polluted in the summer (because as your references suggest it is decoupled from the mid-latitudes) does this not suggest that local sources are unimportant?

*Thank you for pointing this out. Previous work does show that sub-boreal regions can still contribute. We referred to midlatitudes as those in the lower 48, but realize this is not correct. Thus, we simplified the sentence to, "The Arctic summertime atmosphere is historically less polluted as compared to the rest of the year (Quinn et al., 2002; Leaitch et al., 2013; Heintzenberg et al., 2015), thus it is critical to assess the impacts of potentially important local sources of summertime aerosol on Arctic radiation and cloud microphysical processes."*

Page 6 lines 23-24 'Hotspots of larger particles.. were not observed near Prudhoe bay (not shown)' I don't understand why you include this sentence were you expecting to see larger particles?

*This sentence was removed.*

Page 7 lines 2-10 Here you suggest that predominance of larger particles above the BL is the result of growth during vertical transport and dynamical restriction of nucleation mode particles in the BL. It surely can't be the result of both?

*We changed this sentence to say 'or' to demonstrate that it could be from one or the other process.*

Page 8 lines 4-9 beginning 'in terms of indirect forcing' Is the argument here that accumulation mode aerosol derived from nucleation have an impact on cloud properties or that the nucleation mode particles affect cloud directly? If the latter, please explain why marine aerosol may be more hydroscopic then sulphate. Petter et al., (2007, ACP) suggest similar kappa values for both.

*The purpose of this paragraph is to provide broader implications for the direct and indirect impacts of aerosols from this source from previous work. It appears the beginning of that sentence caused confusion, so we removed it. We did not intend to indicate that marine aerosol may be more hygroscopic than sulfate. However, in addition to the sulfate, oil extraction and marine emissions include a host of different organic species that partition to the particle phase, which are not very hygroscopic. Thus, it is difficult to say which general emission source would produce the most hygroscopic aerosol, and is why we broadly state that their hygroscopicities could vary.*

Page 8 line 9-11 beginning 'In general, our results' Given your previous statement that the aerosol appears to have a trivial direct forcing effect and extremely uncertain indirect effect is your final statement really justified?

*This sentence was removed.*

Page 8 lines 27-29 Why would you expect to observe an 'abundance of coarse and accumulation mode particles' if the wildfires generate secondary organic aerosol?

*Although SOA is generally smaller in nature, they can age as they are transported and grow in size. We clarified this here.*

Page 8 line 31 'Which are higher than standard summertime and even springtime haze concentrations' what is the 'standard' concentration and why have you not provided citations here?

*See response to analogous comment by reviewer 1. This part of the sentence was removed.*

Page 9 lines 11-13 'Our observations are parallel to previous summertime observations from regional boreal fires in that they produce substantial quantities of aerosol...' I'm genuinely unsure what you mean by this sentence although I am confident that it is not your observations producing aerosol. Please clarify.

*We changed to 'in that such fires produce substantial quantities of aerosol' to clarify that we meant they are produced from fires.*

Page 11 lines 10-11 ',but demonstrating the larger impact of nucleated aerosol in the vicinity surrounding Prudoe bay' Assuming you mean climate impacts I fail to see how this study demonstrates any impact from these aerosol. You state on page 7 (lines 31-33) that you didn't observe any direct forcing. I also see no evidence (from these observations) of the indirect impact. Please clarify whether the observations reported here do or do not suggest a significant indirect aerosol forcing from emissions in Prudoe bay and provide greater justification for this conclusion.

*We intended to suggest that this is a larger source than previously reported. To reflect this, we changed the sentence to, "Probably the most notable observation was that Prudhoe Bay is a persistent but localized source of black carbon and especially nucleated aerosol, supporting previous findings at Utqiaġvik from Kolesar and colleagues (2017) and Gunsch and colleagues (2017), but demonstrating the larger influence of particle nucleation on the aerosol population in the vicinity surrounding Prudhoe Bay." Additionally, the impacts of these aerosols are detailed by Maahn et al. (2017).*

Page 11 lines 18-21 'With both fire activity and oil exploration projected to increase in a warming climate, these sources will likely continue to make significant contributions...' Previously the authors have stated that emissions in Prudoe bay have a localized impact only, with this in mind can the authors justify so strong a statement on the importance of future oil exploration to Arctic aerosol?

*Even though this source is what we call localized, the HYSPLIT dispersion analyses indicate that dispersion is highest within a 2 degree region, but lower mass concentrations are transported over the entire study area. Thus, a fraction of these particles is transported regionally and could thus have implications for regional climatic impacts. We think this statement is relevant for our results, but have clarified what we mean by 'localized' throughout the manuscript.*

Minor comments
Page 2 line 3 Missing 'the' before climate feedbacks. However, to improve clarity I would suggest replacing 'climate feedback' with 'ice-albedo feedback' (please see first major comment)

*Added 'the' and changed to 'ice-albedo feedback'.*

Page 2 line 6 Use of the phrase 'hinges' is colloquial I would suggest changing to 'depends'

*Done.*

Page 2 line 7 'inherently depends' is a redundancy please delete inherently

*Done.*

Page 2 line 7 Replace 'atmospheric processing' with age

*Changed to 'extent of aging'.*

Page 4 line 9 'important sources of aerosol' You mean important local sources of summertime aerosol?

*Yes, changed to 'local sources of summertime aerosol'.*

Page 4 line 11 To improve clarity please replace 'such' with local

*Done.*

Page 6 line 31- Page 7 line 32 Please split your citations to differentiate between those referencing flaring emissions and those referencing nucleation mechanisms.

*These are already split up.*

Page 7 line 2 Please change 'removal of particles' to 'transfer of particles' or equivalent. Removal suggests removal of the particles from the atmosphere.

*Changed to 'transition of particles'.*

Page 7 line 13 Please cite Stohl et al., 2013 (https://www.atmos-chemphys.net/13/8833/2013/) in reference to BC emissions from flaring.

*Done.*

Page 7 line 27 Please replace 'loss' with 'transition'. The particles aren't lost there just bigger.

*Done.*

Page 8 line 7 Are you referring to diameter or radius here in reference to CCN?

*Clarified that this is referring to diameter.*

Page 8 line 16 'evidenced by the elevated AOD originating from central Alaska' The elevated AOD is the result of aerosol originating from central Alaska. I would suggest rewording to 'the elevated AOD originating from central Alaskan wildfires' or equivalent.

*Changed to 'the elevated AOD originating from the central Alaskan wildfires'.*

Page 8 line 17 'extended until the end of Jul' What was extended until the end of July?

*Changed sentence to, "ACME-V flights were impacted by the high AOD regions from late-Jun until end of Jul."*

Page 10 lines1-2 I agree with this statement but both of the papers cited are concerned only with the Alaskan Arctic. I would also suggest citing Garrett et al., 2010 http://journals.co-action.net/index.php/tellusb/article/view/16525/0, Browse et al., 2012 https://www.atmos-chem-phys.net/12/6775/2012/ or Eckhardt et al., 2003 https://www.atmos-chem-phys.net/3/1769/2003/acp-3-1769-2003.html (among others).

*Thank you for bringing the suggested references to our attention. We have added all three to this sentence.*

---

## Author Response (AR2)

**Reviewer 1**

The manuscript has improved a lot after revision. But I still suggest some comments for minor revisions.

In Figs. 3 and 8, the blue dots are 'Prudhole Bay Boundary layer' (not 'Prudhoe Bay' as shown in the figures) according to the definition in Table2.

*We changed both Figures 3 and 8 captions to be clearer that the figures are showing Prudhoe Bay in the boundary layer.*

P1, L12-14: I think these statements belong to the introduction section.

*We wanted to include some background in the abstract, so we left these statements in the abstract. We touch on these in the introduction already.*

P2, L1-5: please provide citations.

*Done. We included Tsay et al. (1989) for the first sentence and Boucher et al. (2013) to the second sentence. The third sentence is covered by Boucher et al. (2013).*

P5, L4: fossil fuel -> fossil fuel combustion

*Done.*

P6, L26: 'spatial extent of Prudhoe Bay emissions' is confusing. Please revise.

*We changed 'extent' to 'coverage'.*

P8, L29: Summer 2015 season -> Fires in summer 2015

*We changed to 'The 2015 summer fire season'.*

P8, L30: existed -> lasted

*Done.*

P9, L1-2: I suggest add 'during' before every date

*Done.*

P9, L3: 'larger' compared to what?

*We changed 'larger' to 'increased'.*

P9, L10: 'since …' is confusing. Please revise.

*We changed 'since' to 'due to the fact that'.*

P9, L13-14: 'Combined, …' is confusing. Please revise.

*This is referring to the HYSPLIT and MCE data. We revised to clarify.*

P10, L18: the citations should be 'Leaitch et al., 2013; 2016'

*We left as is, since this is the format of the Copernicus Endnote Style downloaded directly from the website.*

P10, L10: 'the presence of … is lower' is wrong. The concentration can be high or low.

*P10, L10 does not contain this information. But we see the reviewer meant P10, L18-19 and revised to say the 'concentrations of pollutants'.*

Section 3.3 used a lot of 'the summer'. If it is general and not referred to as the summer in 2015 discussed in the paper, 'the' should be deleted.

*Done.*

The summary is usually in past tense.

*Most of it already was in past tense, but we changed the second sentence to be past tense.*

**Reviewer 2**

This revised paper accurately reports novel and interesting observations of summertime aerosol from the North Slope of Alaska and I recommend publication. One (very) minor comment concerning a typo is included below.

Minor comments:

In Figure 6: 'Correlations between CO and c) rBC mass and d) scattering efficiencies at 550nm for are also shown' the caption appears to be missing a word after for? It would also improve clarity if the authors defined MCE.

*Typo, we added 'all data' after 'for'. We also defined MCE.*